# Influence of Ball-End Milling Strategy on the Accuracy and Roughness of Free Form Surfaces

Zuzana Grešová [1], Peter Ižol [1], Marek Vrabeľ [2,*], Ľuboš Kaščák [1], Jozef Brindza [2] and Michal Demko [1]

1   Department of Technologies, Materials and Computer Aided Production, Faculty of Mechanical Engineering, Technical University of Košice, Mäsiarska 74, 040 01 Košice, Slovakia; ingzgresova@gmail.com (Z.G.); peter.izol@tuke.sk (P.I.); lubos.kascak@tuke.sk (Ľ.K.); michal.demko@tuke.sk (M.D.)
2   Prototyping and Innovation Centre, Faculty of Mechanical Engineering, Technical University of Košice, Park Komenského 12A, 042 00 Košice, Slovakia; jozef.brindza@tuke.sk
*   Correspondence: marek.vrabel@tuke.sk

**Abstract:** Freeform surfaces are present on an increasing number of engineering products. Three- and multi-axis computer numerical control milling machines are commonly used for improved production. Computer-aided manufacturing (CAM) systems are used almost exclusively for the creation of programs for a variety of machining centers. This study compared the quality of freeform surfaces made by 3- and 5-axis milling using three commonly used strategies (linear, offset, and spiral). The CAM system-predicted surfaces were also compared with the actual surfaces. A test sample with a freeform surface was used for the experiments. Considering the size and distribution, the discrepancy between the predicted surface deviations and the deviations in the produced samples was proven. Maximum negative surface deviations, when 5-axis milling, employed linear and spiral strategy of 29% and 71% less than those produced by the 3-axis milling. On the contrary, positive deviations were 48% smaller. A comparison of the scans showed that the two strategies (linear and spiral) yielded better results for 5-axis milling, and the offset strategy was better for 3-axis milling. Evaluation of the achieved surface roughness showed that the milling method did not significantly affect the surface quality in the linear strategy. However, other two strategies (offset and spiral) achieved better results with 5-axis milling compared to 3-axis milling. The proposed method of evaluating the accuracy of machined free form surfaces can be used in experimental as well as production activities.

**Keywords:** ball-end mill; freeform surfaces; milling strategy; surface deviation; surface roughness

## 1. Introduction

To remain competitive in the current market, the demand to produce increasingly complex high-quality products must be met using efficient production processes. Product variability has increased, and seriality has decreased, which complicates the standardization of processes and the reuse of known technological settings. There are freeform surfaces on an increasing number of engineering products, and computer numerical control (CNC) milling is considered the most efficient, productive, and flexible method for manufacturing these surfaces. The production of freeform surfaces is realized by CNC machines either directly or indirectly. In the first case, the part is made directly on the machine, and in the second case, a forming tool is used (mold, die, and pressing tool), followed by a CNC machine operation. Ball-end mills, owing to their shape, are widely used to finish freeform surfaces; the tool easily adapts to the machining of these types of parts.

Developments in this area are intensive, as evidenced by reviews [1,2]. The first characterizes the situation in the period 1997–2008, and the second analyses the results obtained in the ten years prior to 2021. Reference [1] focuses on three aspects of machining contours: toolpath generation, tool orientation identification, and tool geometry selection. There was noticeable progress in all three areas during the reviewed period, but the authors

also identified the problem areas. They considered the computational and machining times to be the two areas requiring further improvements for efficient machining of freeform surfaces. According to the authors, it is necessary to pay more attention to quality when machining polyhedral, cloud-of-point, compound, and trimmed surfaces. They recommend measuring deviations to evaluate the quality of freeform surface machining. An upper and lower limit should be set. The first controls the maximum scallop height, while the second corresponds to gouging. The authors of a recent study [2] identified gaps for further research in the literature, for example, gaps were identified in the fields of freeform milling of difficult-to-cut materials, tool inclination, three-dimensional (3D) surface roughness, microhardness, and residual stresses. It is also necessary to study atypical milling tools and new types of coolants. The need for continuous improvement in generated toolpaths, simulations, and analytical tools is emphasized.

Milling machines with three or five axes are typically used in the production of parts with free form surfaces. The decision to use a 3- or 5-axis CNC milling machine is not easily made. It determines the complexity and required accuracy of parts, operating costs of machines, productivity, and the required knowledge and experience of programmers and machine operators. To facilitate decision making, the authors of [3] propose a decision-making tool based on fuzzy systems. In general, the following applies to the use of 5-axis machines:

- The production time is reduced by using one fixture and one origin (zero point) instead of several fixtures and zero-point settings for each fixture.
- The accuracy of the parts is higher, in contrast to 3-axis milling, where each loosening and re-clamping has an adverse effect on accuracy.

However, the use of 5-axis machines also has disadvantages:

- The machine price and operating costs are higher than those of a 3-axis machine.
- Staff with a higher level of education and experience are required, whether programmers or operators.
- Programming and additional systems, that is, computer-aided manufacturing (CAM) systems and postprocessors, are more expensive, and the cost of training programmers is also incurred.

Sadílek et al. [4] point out the differences among free form surfaces made using 3-axis, (3 + 2)-axis, and 5-axis milling. It lists the advantages of 5-axis milling over (3 + 2)-axis milling, which include increased milling accuracy, shortened cutting time, extended tool life, increased functional surface properties of the machined surface, and reduced and better orientation of the cutting forces. When comparing the machining accuracy in 3-axis, (3 + 2)-axis, and 5-axis milling, the highest accuracy was achieved in 5-axis simultaneous milling. The (3 + 2)-axis milling yielded more favorable surface roughness values. The comparison was performed on a sample with a shaped surface made of aluminum alloy EN AW 6060. A zig zag (linear) strategy with conventional and climb milling and different tool axis angles was used. The predicted residual material in the CAM system for the group of samples processed using 5-axis milling corresponds closest to the actual residual material. These results are valid for CAM systems. Similar comparisons indicate the quality of computational algorithms used in CAM systems.

Ref. [5] compares 3- and 5-axis milling in machining basic geometric shapes: ball, cylinder, and pyramid. It is stated that both milling methods achieve almost the same shape deviations, but the deviations achieved by 5-axis milling are smaller and the quality of the machined surface is better. Linear and spiral strategies were employed.

A mathematical model for predicting the roughness of a milled surface is defined in [6]. The authors state that after finishing operations with ball-end mills, milled surfaces present two types of inequalities: cusps caused by the cutting width (ae) and scallops caused by the feed per tooth (fz). Kolar et al. [7] describe a virtual machine model that is usable not only in simulations of milling processes, but also for accurate determination of milling time and

surface condition after machining. This model makes it possible to influence the accuracy, quality, and productivity of machining by setting the interpolator.

Surface properties are a significant measure of the quality of the finished component because they affect properties such as the dimensional accuracy, friction coefficient, wear, post-processing requirements, appearance, and cost. Deviations from the required shapes or dimensions are caused by machining-tool errors, improper workpiece clamping and deformation, and worn or improperly clamped tools. The surface roughness is one of the primary requirements for the design of engineering products. It is a widely used product-quality indicator and is usually measured when a component is already machined [8]. Authors present a series of mathematical models for determining the surface roughness ($R_z$) for commonly used shape-milling methods (vertical, push, pull, oblique, oblique push, and oblique pull) and determines the influence of the milling method on the resulting surface roughness in ball-end milling. Aluminum alloy EN AW 7075-T6 was selected as a workpiece material.

The quality of a surface produced by milling also depends on technological parameters, such as cutting conditions and coolants. The evaluation of three cooling methods when machining aluminum alloy EN AW 6061 is described in [9]. The authors conclude that the MQL cooling technique can be used as a better alternative to the dry or flood cooling method.

The machining of freeform surfaces is generally associated with the concept of machining strategy. The milling strategy used depends on the relative positions of the cutting tool and workpiece, as well as the kinematics of the cutting tool during operation. There is increasing emphasis on product quality and process efficiency, which indicates that the industry requires high-efficiency machining strategies for machining freeform surfaces. Therefore, strategies must satisfy the growing demand for surface accuracy and integrity, shorter machining times, and reduced cutting forces.

Reference [10] defined a simulation and optimization system based on body modelling integrated with a CAD/CAM system. Experimental results demonstrated the impact of milling strategies on machining times and their importance in reducing production time and, consequently, costs. Strategies also affect the tool deflection, material removal rate, cutting forces, and machining errors. The linear, concentric and spiral strategies for machining the alloy EN AW 7075 were investigated.

The use of different tool path strategies in ball-end milling on convex, low-curvature surfaces has a significant impact on the cutting forces, surface texture, and machining time [11]. The authors compared linear, offset, spiral and radial strategies. The radial path of the tool provides advantageous results in terms of a more uniform structure and lower cutting forces, but with the highest total machining time. The resulting cutting force is highest when using a helical toolpath owing to the contact area between the tool and chip. Finishing convex surfaces with a small curvature using a helical toolpath strategy is not recommended.

Three strategies, linear, linear rotated by 90° and 3D offset, when machining aluminum alloy are investigated in reference [12]. The 3D offset is the best finishing strategy; it combines the benefits of the linear and linear 90° strategies. Continuous machining strategy is ensured, which is favorable for milling. The tool life is increased, and a better surface quality is achieved.

The impact of the strategies on the surface roughness for die and mold applications was studied in [13]. The path strategy influences the actual machining time, polishing time, and costs. Offset, spiral, radial and zig-zag (linear) strategies were compared. The best results were achieved by the 3D offset and a spiral that slits the part horizontally.

Ramos et al. [14] compared three typical milling strategies when milling a freeform surface: radial, raster, and 3D offset. Their effects on surface roughness, texture, and dimensional deviations were evaluated. The sample was made of aluminum alloy. In this case, the 3D offset strategy proved to be the most suitable because it achieved the most favorable results for each of the monitored parameters. However, according to the authors,

the results cannot be generalized; however, the study shows that there is a general trend when the machining areas are like those tested.

In reference [15], three strategies for high-speed milling were evaluated: linear, linear in two perpendicular directions, and circular. The evaluated parameters were the surface roughness and texture. To achieve the lowest possible surface roughness, a linear path is recommended, and the most uniform surface structure is provided by the circular strategy. The authors recommend avoiding a linear strategy in two perpendicular directions while emphasizing that the results depend on the material being machined.

Considering machining time and average residual load per unit area, [16] evaluated eight finishing strategies for 3- and 5-axis milling. A comparison was made using machine codes that were obtained directly from the CAM system and modified by the optimization software. The authors determined the best strategies for specific types of surfaces while ensuring that feed optimization balanced the machining times of the different strategies. By optimizing the feed rate, the machining time was reduced by 20–50%. The samples were made of aluminum alloy.

Milling strategies were also evaluated based on durability and number of pieces produced, as in the case of a die forging tool [17]. On active tool surfaces, it is advantageous to use a milling strategy with toolpaths that follow the flow of material during the forging process.

The aim of the experiments in the present article was to compare 3- and 5-axis ball-end milling using three machining strategies (linear, offset, spiral), which are commonly used in the production of parts with freeform surfaces. The deviation of the finished shape, the texture of the machined surface and its roughness on a sample specially designed for this purpose were evaluated. The authors of the abovementioned references used an optical microscope to evaluate the deviations of the machined surface. In two cases, a coordinate measuring machine was used. The present study utilizes a 3D laser scanner for this purpose. Within a research article comparing 3-axis and 5-axis milling, the operations are defined separately. In the described experiment, the function of converting 3-axis operations to 5-axes was deployed. This ensured the match of the parameters of the assessed strategies.

## 2. Research Methodology

To achieve the set goal, the experiment was carried out in the following steps:

1.  Deviations in freeform surfaces were detected using a CAM prediction software and a scan of the actual surfaces. The degree of conformity and reliability of the CAM system were compared to predict the accuracy of the manufactured surfaces. Differences between the surface deviations made by 3- and 5-axis milling were obtained.
2.  A comparison of surface textures was made using both milling methods. The areas were evaluated under a microscope, and the differences between the 3- and 5-axis milling surfaces were determined.
3.  Measurements were obtained, and a comparison of the surface roughness of machined surfaces using both milling methods was made.

A sample with a freely modeled 3D surface was designed for the experiment. To design a test sample, the SolidWorks CAD system and its tool, called Freeform, was deployed. The sample (Figure 1) had floor plan dimensions of 50 mm × 50 mm, with a height of 19 mm. The workpiece material of the samples was aluminum alloy, EN AW 6061 T651. It is an Al–Mg–Si–Cu alloy for highly loaded structural applications with tensile strength 260 MPa. It is used in the production of blow molds, compression molds and low volume injection molds. The initial dimensions of the semi-finished product were 50 mm × 50 mm × 20 mm.

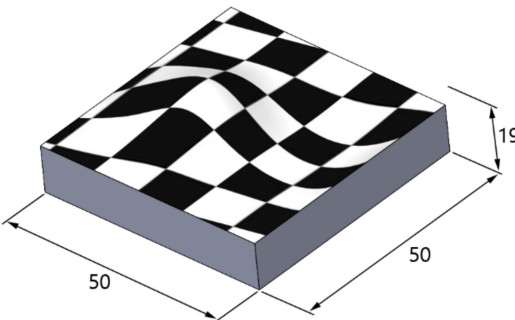

**Figure 1.** Model of the proposed sample with freeform surface.

A CAM system, SolidCAM 2021, was selected to create the NC program. A 5-axis continuous milling machine DMU 60 eVo (DMG Mori Corporation, Bielefeld, Germany) with a control system Heidenhain TNC 640 was used to fabricate the samples for both milling methods. The maximum spindle speed of the machine is 20,000 rpm. The control system has a Look Ahead function which detects changes in the direction of the milling tool in advance [18]. The system adjusts the tool feed rate based on the tolerance entered in the respective cycle. Related cycle 32 Tolerance was not used so as not to affect the results of NC programs obtained from the CAM system.

A KSX 125 vise (supplier Schunk GmbH & Co. KG, Brackenheim, Germany) with a set clamping force of 25 kN was used to clamp the sample. The sample was clamped in the middle of the vise's jaws. Each sample was prepared in six operations. The first five operations focused on roughing and pre-finishing the shape. These operations were matched for all samples. After pre-finishing, an allowance of 0.25 mm remained on the surface. The last (sixth) operation was finishing, which was sample specific. As shown in Figure 2, the surface of the sample was prepared for the finishing operation.

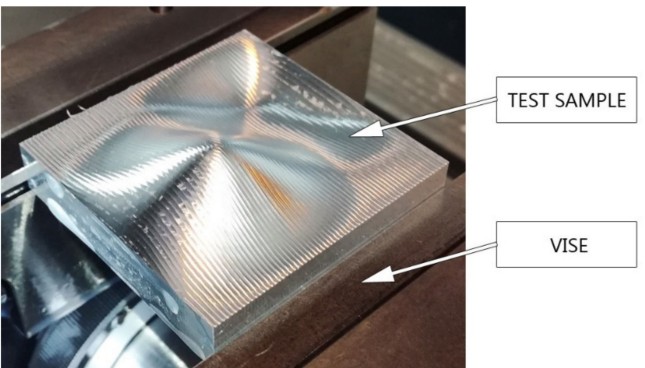

**Figure 2.** Sample surface after roughing and pre-finishing operations.

Three types of finishing strategies were selected for the comparison: linear, offset, and spiral. In the linear strategy, the tool transitions are parallel in the XY plane and follow the surface in the Z-axis direction. Using this strategy, the direction of the toolpaths can be selected. To shorten the machining time, a direction parallel to the longer side of the workpiece is typically chosen. In offset milling, toolpaths follow the circumference of the machined surface. The path can start at the circumference and progress to the middle surface, or vice versa. The spiral strategy creates a helical path for the tool from a given focus while maintaining a constant contact between the tool and workpiece. Each strategy was applied in 3- and 5-axis milling. The "convert" function of the CAM system was used to program the existing 3-axis operation to a 5-axis milling operation. This approach ensured that most of the parameters of the compared strategies were matched.

The ball-end mill N.RD.10,0.45°.Z4.HA.K TI1000 (manufactured by WNT, supplier Ceratizit S.A., Mamer, Luxembourg) with a diameter of 10 mm was used to finish the

freeform surfaces. Parameters of the tool are listed in Table 1. The tool operated at a cutting speed of $v_c = 400$ m.min$^{-1}$ and a feed per tooth of fz = 0.04 mm. The parameters correspond to the tool manufacturer's recommendations for the selected aluminum alloy. The same parameters were used in the settings of all assessed strategies, and the main requirement was the height of the cusp after machining (scallop height), which was set to 0.01 mm. All the tests were realized with 8% emulsion Zubora 65 H Extra (Zeller + Gmelin GmbH & Co. KG, Eislingen/Fils, Germany). This cutting fluid is designed for machining steel, cast iron as well as aluminum. The flood cooling method was deployed.

**Table 1.** Parameters of the ball-end mill tool [19].

| ød$_1$ [mm] | ød$_1$ [mm] | l1 [mm] | l$_2$ [mm] | l$_3$ [mm] | z [-] | $\gamma_s$ [°] | $\lambda_s$ [°] |
|---|---|---|---|---|---|---|---|
| 10 | 10 | 14 | 27 | 67 | 4 | 15 | 45 |
| Sketch of the tool | | | | | | | |

### 3. Results

#### 3.1. Surface Deviations

The first step was to compare the surface deviations predicted by the CAM system with the real deviations obtained using 3D laser scanning. The Solid Verify simulation mode was used in the CAM system to predict the surface; Handy SCAN BLACK Elite scanner (Creaform Corporation, Levis, Canada) was used to scan the samples; and VXelements Viewer software [20] was used to evaluate the scans. The software compares the scan to a 3D CAD model that must be loaded into the software.

Color scales with the assigned deviation values for the CAM and VXelement Viewer evaluation software are shown in Figures 3 and 4, respectively. For negative deviations, the scale was set identically, and for positive deviations, a maximum value of 0.01 mm was entered for the CAM system. The reason for this is that a limited number of locations were used to define the scale. This did not distort the comparison because the CAM system did not show positive deviations above this value.

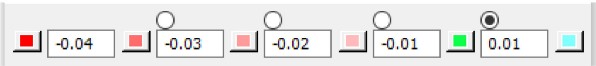

**Figure 3.** Color scale with deviation values for the CAM system.

Table 2 compares the results of simulated sample machining using a CAM system and the scanning results of actual produced samples for 3-axis milling. The CAM system presented the virtually machined surfaces with negative deviations only, and deviations in the positive direction were not presented. Minimal deviations were obtained with the linear strategy, and the negative deviations in other strategies exceeded −0.04 mm. Scanning of the actual surfaces showed deviations in both the positive and negative directions. The extreme values of the scale (−0.05 mm and +0.04 mm) were reached. In all the strategies, the deviations were distributed almost equally. Edges of the surface parallel to the X-axis of the reference coordinate system had a positive deviation. The side of the test sample peak toward the valley had negative deviations (red areas on the scanned surfaces). The CAM system did not correctly predict the condition of the finished surfaces.

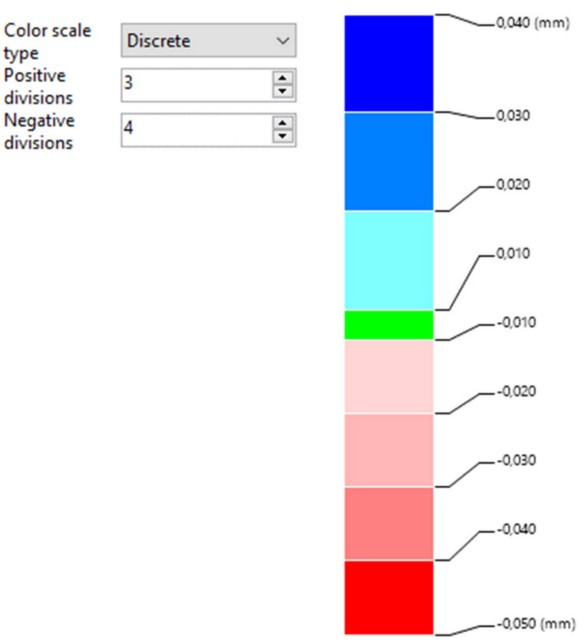

**Figure 4.** Color scale and deviation values for the scan evaluation software.

**Table 2.** Comparison of simulated and laser scanned surfaces of samples made by 3-axis milling.

| Strategy | CAM Simulated Surface | Laser Scanned Surface |
|---|---|---|
| Linear | | |
| Offset | | |

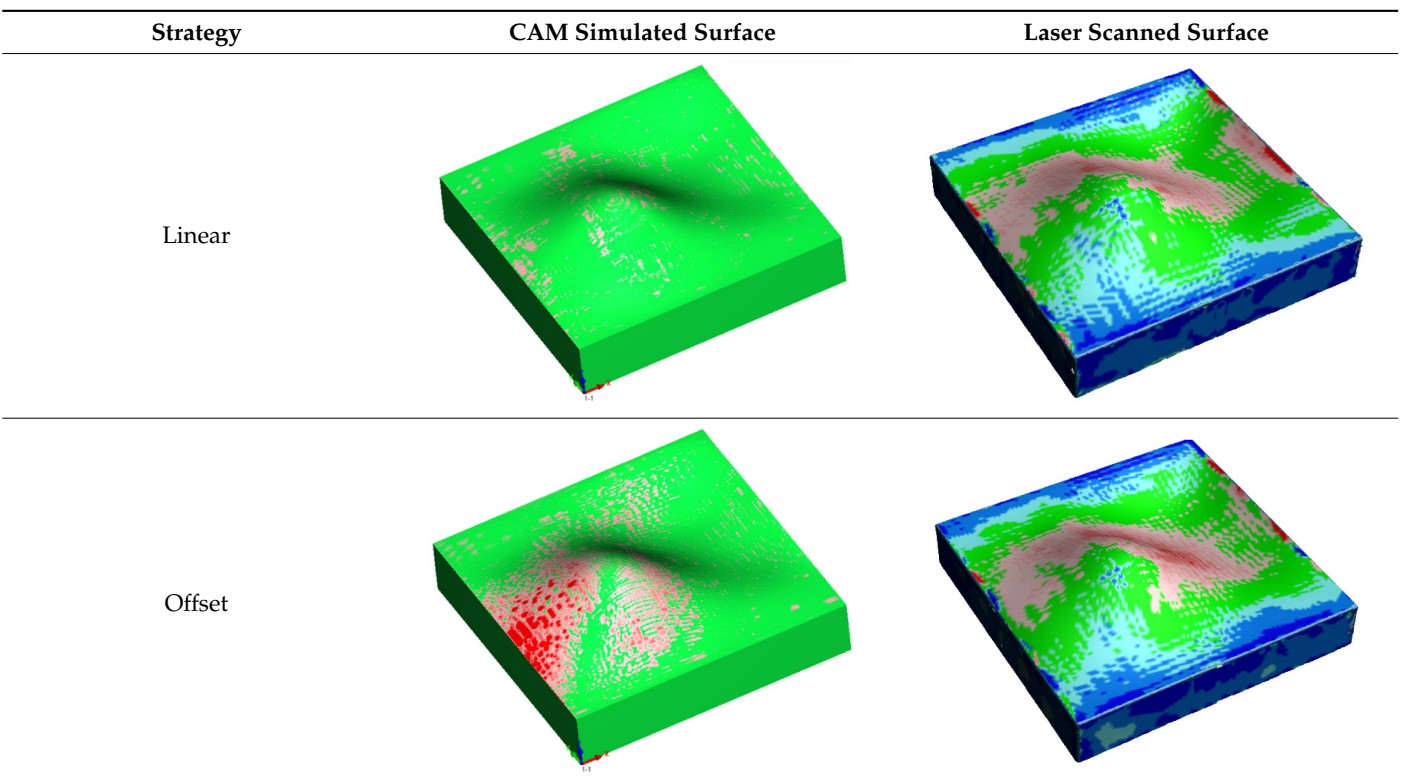

**Table 2.** *Cont.*

| Strategy | CAM Simulated Surface | Laser Scanned Surface |
|---|---|---|
| Spiral | 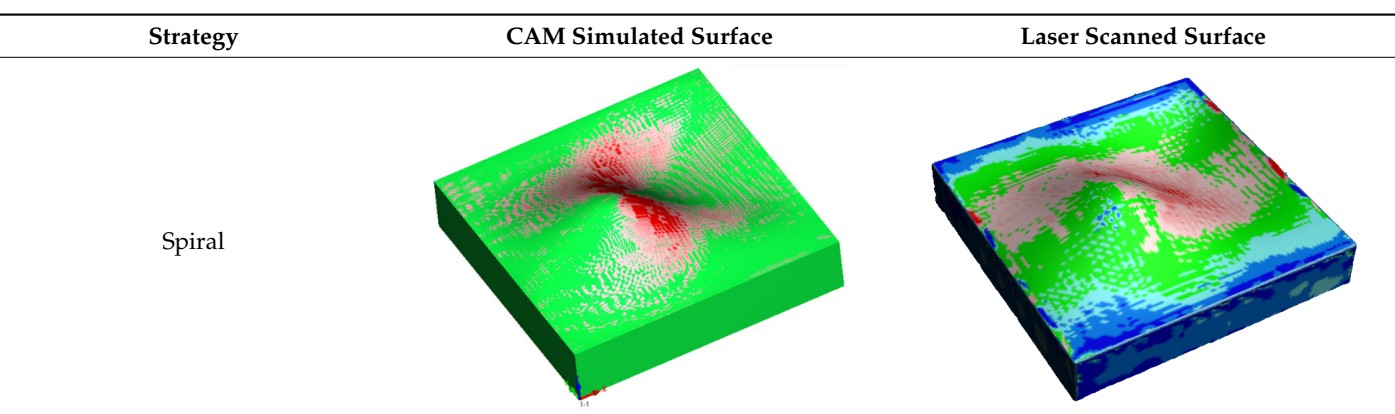 | |

Table 3 lists the predicted and scanned surfaces for 5-axis milling, realized by 3-axis milling conversion. Even in this case, the predicted distribution of deviations did not coincide with the real distribution. A partial agreement was reached using the linear strategy. At the output of the CAM system, there are otherwise distributed areas with negative deviations. Negative deviations with a limit of −0.04 mm were predicted for the spiral strategy. As an output from CAM system, distributed areas with negative deviations were noticeable and highest negative deviations with a limit of −0.04 mm were predicted for the spiral strategy. On the real surfaces, these negative deviations occurred on the offset sample. For this case, areas with positive deviations up to +0.04 mm occurred at the edges of the surface parallel to the X-axis.

**Table 3.** Comparison of simulated and laser scanned surfaces of samples made by 5-axis milling.

| Strategy | CAM Simulated Surface | Laser Scanned Surface |
|---|---|---|
| Linear | 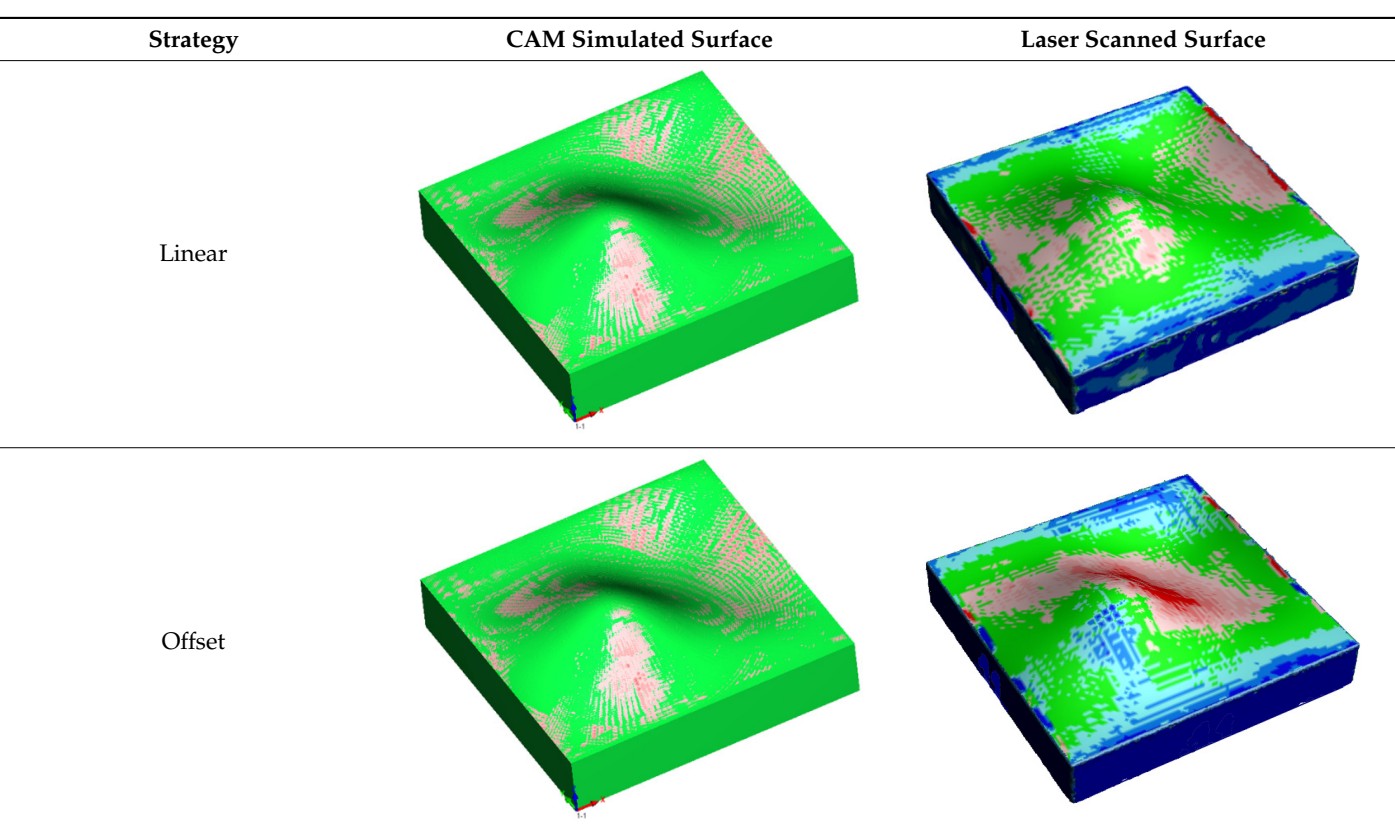 | |
| Offset | | |

**Table 3.** *Cont.*

| Strategy | CAM Simulated Surface | Laser Scanned Surface |
| --- | --- | --- |
| Spiral | 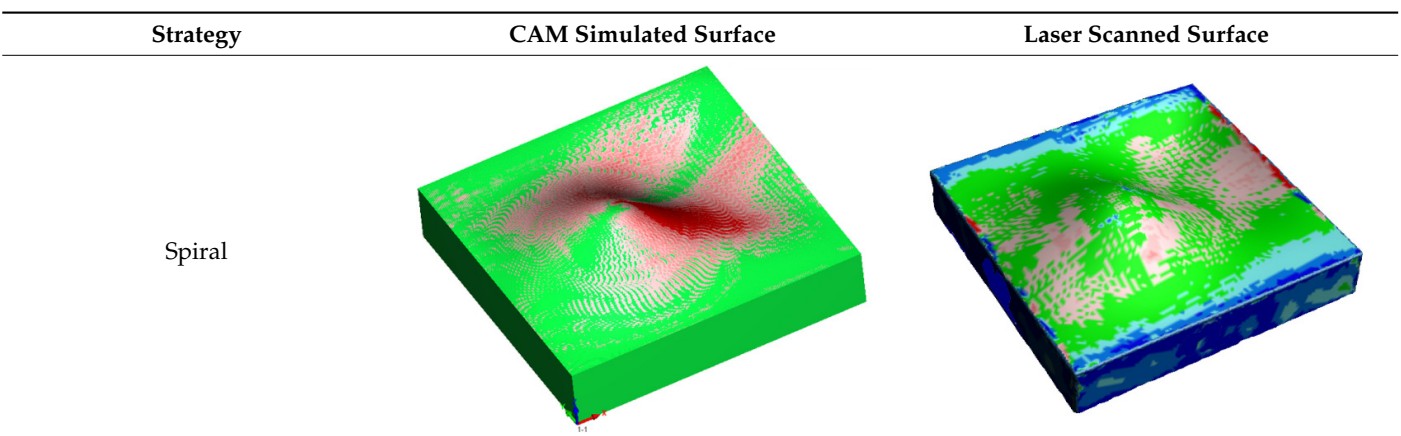 | |

In addition to analysis using color maps, the ability to directly display deviation values on the sample surface was used to evaluate the data obtained by scanning. The VXelements Viewer software allows the translation of a surface grid in which nodal points display specific deviation values. It is still possible to insert points into the required location, at which the deviation values are also displayed. In this way, the values obtained for plotting are the average deviations and maximum values of the negative deviations. An example of displaying the numerical values of deviations is shown in Figure 5, where the scanned surfaces of the 3- and 5-axis spiral milling strategies are compared.

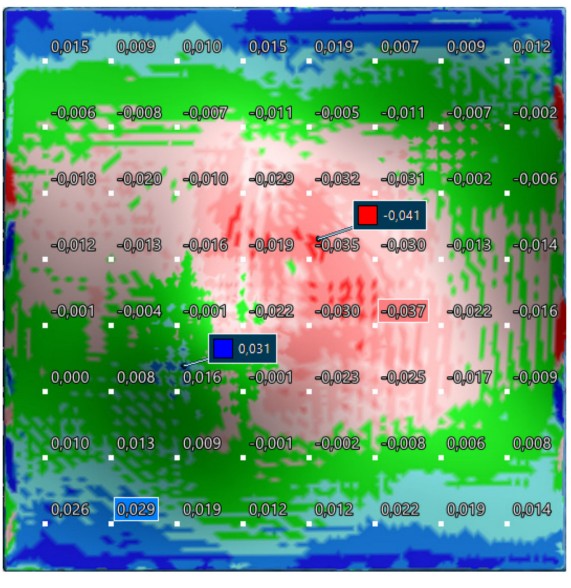

(**a**)

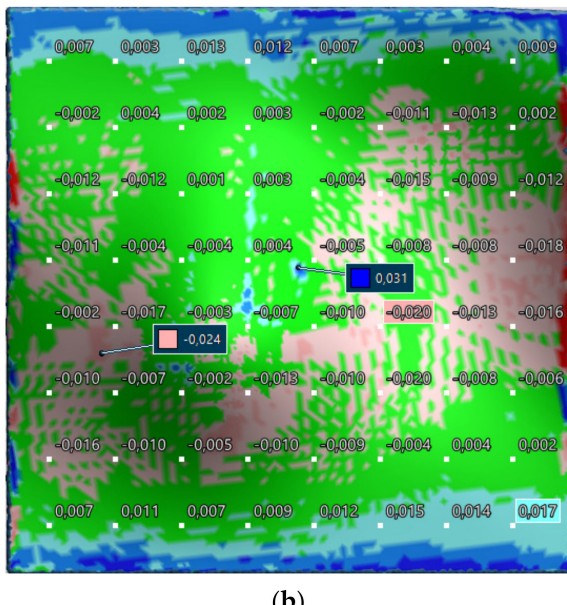

(**b**)

**Figure 5.** Deviation values for scanned sample surfaces made using the spiral strategy: (**a**) 3- and (**b**) 5-axis converted milling.

Table 4 shows the average and the maximum values of positive and negative deviations for all machined samples.

Based on the data given in the previous table, a graph of the average deviations and maximum values of the positive and negative deviations for each sample is shown in Figure 6. For 3-axis milling, the largest average negative deviation was achieved by the spiral strategy, and the largest positive deviation achieved by the linear strategy. The largest positive and negative deviations in 5-axis milling were obtained by the offset strategy.

**Table 4.** Deviation values [mm] for samples made by 3-axis and 5-axis milling.

| Value | 3-Axis Milling | | | 5-Axis Milling | | |
|---|---|---|---|---|---|---|
| | Linear | Offset | Spiral | Linear | Offset | Spiral |
| Average | 0.002 | 0.000 | −0.004 | −0.001 | −0.005 | −0.003 |
| Negative deviation | −0.040 | −0.042 | −0.041 | −0.031 | −0.049 | −0.024 |
| Positive deviation | 0.034 | 0.028 | 0.031 | 0.023 | 0.037 | 0.031 |

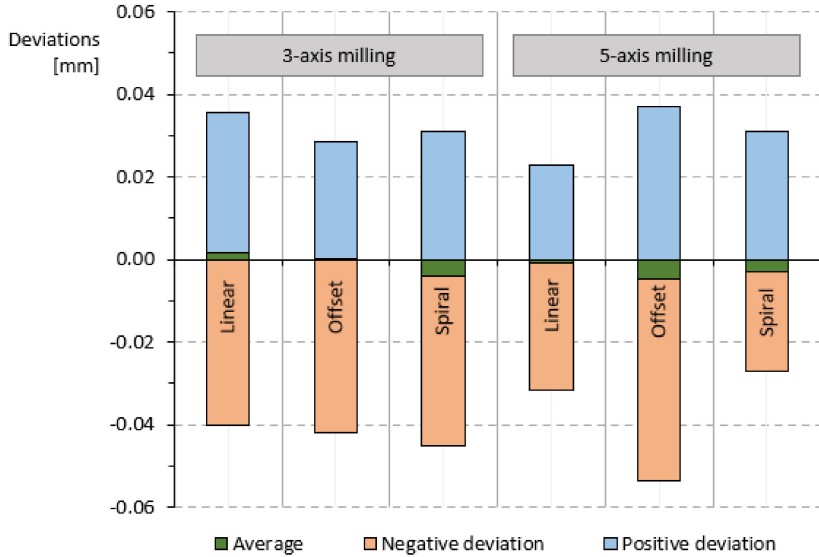

**Figure 6.** Graph of the average deviation, maximum negative and positive deviation for each sample.

One of the objectives of the study was to compare the accuracy of the surfaces made using 3- and 5-axis milling. Five-axis milling obtained a better result with the linear and spiral strategies, and most of the freeform areas showed no significant deviations. In contrast, for the offset strategy, 5-axis milling obtained the worst result; negative deviation at −0.04 mm was noticeable on a larger part of the area obtained using 5-axis milling than that obtained using 3-axis milling.

### 3.2. Surface Texture

The surfaces of the machined samples were examined using a VHX-5000 microscope (Keyence Corporation, Osaka, Japan). Three areas were selected on the test sample with a freeform surface: the planar area (1), the area on the inclined wall of the protrusion (2), and the area on the inclined toward to valley (3) (Figure 7). Photographs at 200× magnification were taken at these locations. The bottom and left sides of each photograph were oriented in the X-and Y-axis directions, respectively, which corresponded to the positions at which the samples were produced.

The surface roughness was also measured at the locations where the microscope photographs were taken. Measurements were performed using a portable surface roughness tester Surf test SJ 410 (Mitutoyo Corporation, Headquarters: Takatsu-ku, Kawasaki City, Japan). During the measurement, the probe tip moved perpendicular to the direction of the tool paths at a given location. A cut-off value of 0.8 mm was selected, as recommended by the ISO 4287:1997 standard. Measurements were performed three times on each surface, and each presented value is the average of these three measurements. The values of parameters $R_a$ (arithmetical mean roughness), $R_z$ (mean roughness depth), $R_t$ (maximum height of the profile), and $R_q$ (root mean square of the surface roughness) were recorded.

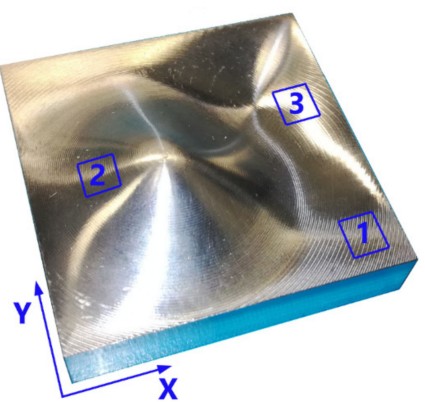

**Figure 7.** Sample with marked locations for microscopic scanning and surface roughness measurement.

Table 5 shows a surface image of the samples made using 3-axis milling. At location 1, there are clear traces of a tool, where, owing to its position relative to this surface element, the cutting speed near the tool's tip is equal to zero. The material was crushed or plowed in this location. In locations 2 and 3, the tool occupied a larger diameter of the hemispherical section, which is evidenced by the regularly arranged traces of the tool. Because of the larger foot radius at location 2, it can be concluded that in this area, the tool worked with a larger radius than that in location 3.

**Table 5.** Surface texture and surface roughness in 3-axis milling.

| Strategy | Shooting Location No. 1 | Shooting Location No. 2 | Shooting Location No. 3 |
|---|---|---|---|
| Linear | | | |
| $R_a$ [μm] | 0.89 | 0.95 | 0.77 |
| $R_z$ [μm] | 4.76 | 3.93 | 3.56 |
| $R_t$ [μm] | 5.70 | 4.8 | 4.21 |
| $R_q$ [μm] | 1.06 | 1.16 | 0.92 |
| Offset | | | |
| $R_a$ [μm] | 0.93 | 0.66 | 0.85 |
| $R_z$ [μm] | 5.04 | 3.45 | 4.42 |
| $R_t$ [μm] | 5.75 | 3.94 | 6.42 |
| $R_q$ [μm] | 0.79 | 0.79 | 1.01 |

**Table 5.** *Cont.*

| Strategy | Shooting Location No. 1 | Shooting Location No. 2 | Shooting Location No. 3 |
|---|---|---|---|
| Spiral |  |  |  |
| $R_a$ [μm] | 1.00 | 0.76 | 1.00 |
| $R_z$ [μm] | 5.49 | 3.73 | 4.65 |
| $R_t$ [μm] | 6.64 | 4.22 | 5.07 |
| $R_q$ [μm] | 1.20 | 4.65 | 1.16 |

The table also lists the surface roughness values. The evaluation is presented in the following section.

The surfaces of the samples made using 5-axis milling are shown in Table 6. A significant difference is observed in the planar parts of the samples. By tilting the tool, contact with the tool area with zero cutting speed was eliminated and traces of the tool were regular, similar to those of other sample areas.

**Table 6.** Surface texture and surface roughness in 5-axis milling.

| Strategy | Shooting Location No. 1 | Shooting Location No. 2 | Shooting Location No. 3 |
|---|---|---|---|
| Linear |  |  |  |
| $R_a$ [μm] | 0.94 | 0.92 | 0.75 |
| $R_z$ [μm] | 3.94 | 4.31 | 4.01 |
| $R_t$ [μm] | 4.72 | 5.69 | 4.72 |
| $R_q$ [μm] | 1.07 | 1.13 | 0.91 |

**Table 6.** *Cont.*

| Strategy | Shooting Location No. 1 | Shooting Location No. 2 | Shooting Location No. 3 |
|---|---|---|---|
| Offset |  |  |  |
| $R_a$ [μm] | 0.64 | 0.69 | 0.84 |
| $R_z$ [μm] | 3.53 | 3.53 | 4.04 |
| $R_t$ [μm] | 4.00 | 4.46 | 5.28 |
| $R_q$ [μm] | 0.78 | 0.93 | 1.00 |
| Spiral |  |  |  |
| $R_a$ [μm] | 0.63 | 0.76 | 0.94 |
| $R_z$ [μm] | 3.39 | 4.07 | 4.54 |
| $R_t$ [μm] | 3.80 | 4.96 | 4.96 |
| $R_q$ [μm] | 0.76 | 0.91 | 1.11 |

### 3.3. Surface Roughness

Surface roughness parameters $R_a$, $R_z$, $R_q$, and $R_t$ were measured perpendicular to the toolpaths. The measurements were performed at three locations on each sample. The resulting values used in the following graphs correspond to the mean values of the data obtained. Figure 8 compares the surface roughness of the samples made by 3- and 5-axis milling using the commonly evaluated parameters ($R_a$ and $R_z$). For the linear strategy, the values were the same, and for the offset and spiral strategies, the surfaces made by 5-axis milling had a lower surface roughness value for both parameters considered. For 5-axis milling, the offset strategy yielded the best result.

A comparison of the surface roughness of the samples made by 3- and 5-axis milling using parameters $R_t$ and $R_q$ is shown in Figure 9. In the linear strategy, the values are very similar. In the offset and spiral strategies, the surfaces made by 5-axis milling show lower values for both surface roughness parameters. Even for this case, the offset strategy showed the best result in 5-axis milling.

These comparisons show that the linear strategy obtained almost similar surface roughness properties regardless of the milling method. Considering the investigated surface roughness parameters, the offset and spiral strategies were proven to be advantageous for 5-axis milling.

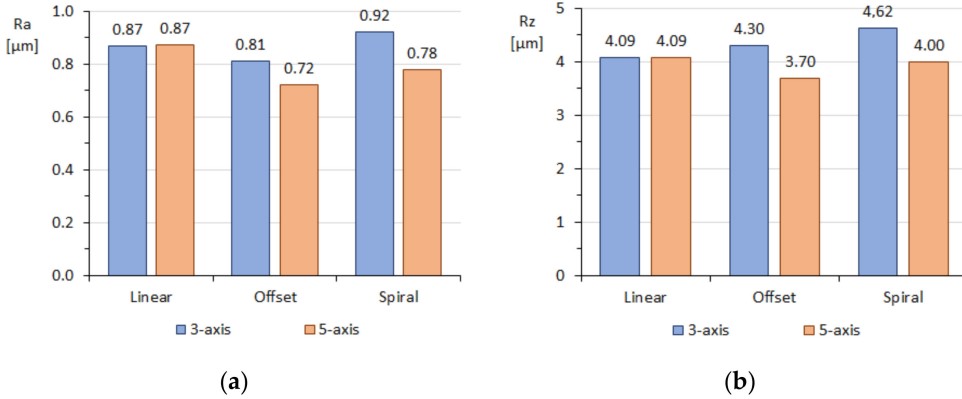

**Figure 8.** Comparison of surface roughness for the strategies performed using 3- and 5-axis milling: (**a**) comparison of $R_a$ and (**b**) $R_z$ parameter values.

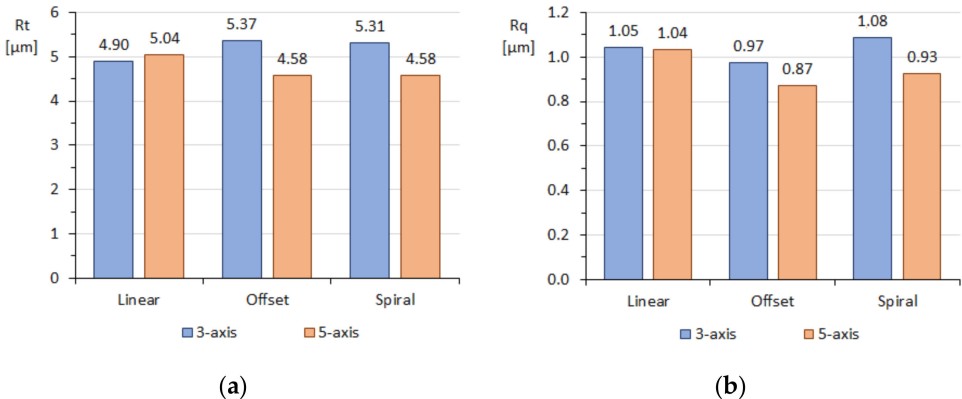

**Figure 9.** Comparison of the surface roughness for the strategies evaluated using 3- and 5-axis milling: (**a**) comparison of $R_t$ and (**b**) $R_q$ parameter values.

## 4. Results and Discussion

The predicted positive and negative deviations in the CAM system did not correspond to the actual values. Authors Sadílek et al. [4] stated that the CAM prediction for 5-axis milling corresponded better to the actual state than that for 3-axis milling. In our case, it was not possible to determine the best or worst match rate. In all the cases, the CAM system predicted only negative deviations. The areas with negative deviations were differently distributed on the actual samples, and areas with residual material also occurred. Positive deviations mainly occurred at the two opposite edges of the sample. Because the specimens were clamped in these places, it is probable that they deformed during machining, owing to clamping forces.

The authors of [4] present positive deviations with a maximum value of 42 μm in 3-axis milling, and deviations in the positive region with an arithmetic diameter of 18 μm in 5-axis milling. In our case, positive and negative deviations were achieved in both milling methods. In the 3-axis milling, a negative maximum deviation of 42 μm was obtained. In 5-axis milling, the arithmetic mean of the negative deviations was 35 μm and the positive deviations 30 μm. One of the reasons for the different results may be the use of Cycle 32 Tolerance of the machine control system described by the authors [4].

Predicting the surface after milling using a CAM system is not sufficient; the results are merely indicative. As stated by the authors in ref. [21], the results obtained in this way cannot be generalized because each CAM system works with a different mathematical model, and the results, especially for 5-axis milling, depend on the setting of a number of parameters. However, such predictions are necessary because they allow CNC program-

mers to evaluate the milling process better when selecting an effective toolpath strategy and milling method.

A comparison of the scans showed that the two strategies (linear and spiral) yielded better results for 5-axis milling, and the offset strategy was better for 3-axis milling. The linear strategy and 5-axis milling achieved a maximum negative deviation of 29% less and a maximum positive deviation of 48% less than the 3-axis milling. With the spiral strategy, the maximum negative deviation was 71% smaller, the positive maximum deviations were the same. With the offset strategy and 5-axis milling, the maximum negative deviation was 17% greater and the maximum positive deviation was 32% greater than with 3-axis milling. This partially confirms the results reported previously [5].

Finishing operations account for 70–80% of the total machining time when making die and mold tools [6]. Surface roughness and surface texture are key indicators of the machined surface quality. Therefore, it is important to optimize the relationship between the milling parameters and these finished part indicators. This eliminates or minimizes finishing operations, such as grinding or polishing, respectively.

When comparing the surface textures, a more uniform surface was obtained by 5-axis milling. The most significant is the difference in the planar parts of the surface. Circular traces left by the cutting-edge contact point, which rotates at a small radius, are visible on 3-axis milling samples. On samples machined by 5-axis milling, tool traces are even over the entire surface. On the textures in [5], the tracks are wider, which is caused by a fixed radial depth of cut. In our case, the scallop height value was set, and the CAM system adjusted the distances between the toolpaths accordingly.

Evaluation of the achieved surface roughness showed that the milling method did not significantly affect the surface quality in the linear strategy. The other two strategies achieved better results with 5-axis milling, compared to 3-axis milling. The offset strategy improved the surface by 13% at the parameter $R_a$ and by 16% at the parameter $R_z$. At the spiral strategy, there was an improvement of 18% resp. 16%. Similar values were obtained for the parameters $R_t$ and $R_q$. The 5-axis milling improved the surface with the offset strategy by 17% resp. 12%, with a spiral strategy for both parameters by 16%. This confirms the results obtained in [4,5]. The offset strategy exhibited the best result for 5-axis milling, which corresponds to the results reported in [12,14].

The surface roughness depends on many parameters. The parameters with the greatest influence are the stepover and feed per tooth; the position of the tool relative to the machined surface is also important. According to [22], 5-axis milling results in a more favorable surface quality and a smaller and more balanced value of parameter $R_z$ owing to the constant position of the tool relative to the machined surface. This finding was confirmed experimentally.

It should be noted that the results obtained from similar experiments depend on many decisions and settings, including the sample material, chosen cutting tool, selected strategies, machine tool, programming method, and programming system. However, this study may provide a general trend in the surface treatment quality for shapes similar to those tested.

## 5. Conclusions

In this study, the milling of freeform surfaces using a ball-end mill, programmed using a CAM system, was experimentally investigated. Three milling strategies (linear, offset, and spiral) and two milling methods (3- and 5-axis) were compared. The size and distribution of the deviations simulated using a CAM system were inaccurate. A comparison of textures showed that 5-axis milling is advantageous in terms of the regularity of tool marks on the finished surface. This is important in the final finishing if the surface is located on a forming tool (mold, die, or pressing tool). A comparison of the surface roughness also shows the advantages of 5-axis milling, which achieved identical or better results than 3-axis milling.

The results obtained complement the knowledge about milling freeform surfaces. The quality of the data acquired by a 3D laser scanner proved the validity of its use. The laser

scanner allows to quickly capture the desired surface with high accuracy and save the data (point cloud) as a 3D model. Due to this option, evaluation software for 3D models allows the results to be presented in several ways. The advantages of the 3D surface mapping compared to the 2D techniques provided by other evaluation devices are obvious. Based on the reviewed literature, it can be stated that the proposed method of verifying the accuracy of the production of freeform surfaces brings a new perspective on this issue. The improving accessibility of these devices, high level of operability and rapid data evaluation, contribute for their rapid expansion in both experimental and practical activities. The availability of the conversion function in the programming system greatly simplifies the preparation of 5-axis operations.

The aim of future work is to design and verify other methods for detecting shape and surface integrity deviations on finished parts. The results obtained can then be compared with the results presented in this article.

**Author Contributions:** Conceptualization, P.I. and Z.G.; methodology, P.I.; software, Ľ.K., J.B. and M.D.; validation, M.V., P.I. and Z.G.; formal analysis, Z.G.; investigation, Ľ.K. and J.B.; resources, P.I.; data curation, Z.G.; writing—original draft preparation, P.I. and Z.G.; writing—review and editing, M.V.; visualization, P.I. and M.D.; supervision, M.V.; project administration, P.I.; funding acquisition, M.V. All authors have read and agreed to the published version of the manuscript.

**Funding:** This research was funded by the Scientific Grant Agency of the Ministry of Education of the Slovak Republic (projects VEGA 1/0457/21 and KEGA 048TUKE-4/2020).

**Institutional Review Board Statement:** Not applicable.

**Informed Consent Statement:** Not applicable.

**Data Availability Statement:** The data presented in this study are available on request from the corresponding author.

**Conflicts of Interest:** The authors declare no conflict of interest. The funders had no role in the design of the study; in the collection, analyses, or interpretation of data; in the writing of the manuscript, or in the decision to publish the results.

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
