# Peer review of "Influence of Ball-End Milling Strategy on the Accuracy and Roughness of Free Form Surfaces"

_applsci, doi:10.3390/app12094421_

Round 1
Reviewer 1 Report
Dear authors,
This manuscript titled " Influence of ball-end milling strategy on the accuracy and 2 roughness of free form surfaces" is focused on an interesting research topic that has been extensively studied in the last years. Thereby, the novelties/contributions of the proposed work should be explained clearly before considering the paper for publication.
The paper has a good technical content and an adequate methodology but presents some important deficiencies in its conception and also in its form. The main concern is the scientific impact and the novelty of the paper. Since the paper could provide a detailed case study analysis rather that providing new scientific knowledge, in the real practise, an evaluation of the results regarding the design to be manufactured and not only regarding the predictive models obtained from the CAM simulation would be necessary. The authors must consider improving the manuscript in the following terms before the paper can be published.
These and other questions are highlighted in yellow in the revised PDF and including notes to take in consideration for a new version of the manuscript.
1.- The cited references must be related to the conditions and objectives of this work. Please justify the relationship between each of the cited references and this work (interest of the material, justification of the technological conditions set, type of tool used, novelties/differences in the methodology, justification of surface milling strategies, goals, etc).
The main contributions of the work from the point of view of research objectives must be addressed in this final paragraph of the Introduction.
The contributions and novelties that differentiate this work from the previous works cited in the Introduction are not clear.
2.- The use/interest/applications of the machined alloy mut be justified adequately.
3.- ¿Which are the main characteristically programming functions of this CNC Model/Brand that could affect the shape accuracy and surface quality of the 3-5 simultaneous milling works? ¿ Does this CNC Model/Brand have/work with any look ahead function? Describe the capacities of the used CNC equipment in this sense.
Cite/reference and include between ( ) the supplier information.
4.- It is convenient to give a detailed description of the milling tool used. Maybe giving reference of the supplier’s catalogue and/or adding a figure/table with all the specific characteristics of the tool used.
5.- Cite all the supplier data reference between parenthesis and/or webpage citation included in references (Catalogue, Supplier WNT, City, Country) [webpage ref.].
6.- The technological conditions used in the experiments must be justified adequately.
7.- Why was the designed model not considered in the comparative analysis and not only the result predicted by the SolidCAM simulator against the manufactured (scanned) result?
These results should be included in the comparison, that is: final result (scanned) vs. model designed (to be manufactured).
8.- Cite supplier reference between parenthesis and/or webpage citation included in references for VX software.
9.- The comparison between the results of the present work and the results of the consulted previous works deserves a more detailed discussion. Please, give a discussion comparing effective values of the output variable object of study in the case of ref. [5] and referred to the surface textures.
It should be considered that texture analysis performed only on 2D images (photographs) is not the most adequate method to evaluate and/or compare this characteristic. In this sense, confocal microscopy is a surface metrology system that has become an extremely important field of science and engineering and surface topography characterization (waviness, roughness, lay of the surface, etc), that plays a crucial role in determining the mechanical, thermal, optical, and/or electrical properties of materials which are used for many modern technologies, components, parts, and products, such as motors, coatings, electronic devices, etc. Thus, the authors should consider a topographical analysis performed using this type of technology instead of 2D photos.
10.- In terms of the surface roughness results, the discussion about the results of the present work compared with the results of the consulted previous works is vague and deserves a more detailed discussion. Please, give a discussion comparing effective values of the output variable object of study.
11.- The results of a technical application work like this, and especially in terms of the quality of the surface finish, depend largely on the system and the type of lubrication/cooling applied in the cutting/tool area. No reference is made to this very important aspect in any part of the work, nor are these conditions specified in the Materials and Methods section.
The work in this aspect should be improved, from the Introduction: there are no citations on works that assess the most appropriate system/type of lubricant in this type of work/alloy.
Once the conditions implemented in this work have been described (system, conditions and type of lubricant; composition, supplier, etc.) and regarding this issue, these must be conveniently justified in the Materials and Methods section.
The Discussion section should also include how this question has affected the results of this specific work and differences/similarities with respect to the results obtained in other works.
Reviewer 2 Report
The paper presented to review corresponds to the Journal’s topics. It has novel results and a practical value to the metalworking industry. The paper is well-prepared and visualized.
In my opinion, the paper has several drawbacks and needs minor revision.
- The paper has the following sections: 1 Introduction, 2 Experimental details, 3 Discussion, 4 Conclusions. However, it will be reasonable to restructure a manuscript by dividing section “2 Experimental studies” into “2 Research Methodology” and “3 Results”.
- The aim of the research should be specified in the Introduction. It will help a reader with the necessity of the literature review analysis.
- The literature review in the “1 Introduction” needs to be restructured according to the parameters impacting the milling strategy selection (e.g., time/cost, cutting modes, accuracy, productivity, etc.).
- Figure 2 describes the test sample fixing during machining. It will be reasonable to admit the selected locating chart and clamping force used.
- How many samples were used for experimental studies (for 3 strategies and 2 methods)? How many tests did you complete to interpret the results adequately?
- The manuscript did not specify the metalworking equipment for 3-axis and 5-axis machining during experimental studies.
- Absolute values of the research results were presented in the main part of the article. In the conclusions, giving relative indicators (in percentage) is desirable to prove the efficiency of the proposed solutions.
- The scientific novelty of the research should be highlighted in the Conclusions. What are the main scientific results covering the research gap of the previous studies?
Round 2
Reviewer 1 Report
Dear authors,
I advised that you have considered all the most significant suggestions in the new version of the manuscript but, in my opinion, the main contributions and novelties that differentiate this work from the previous works cited in the Introduction are not clear yet.
You should bear in mind that this work contributes little from a scientific point of view and, as a technical report, the new points in terms of technique and compared to previous similar works should be very clear.
To reinforce this lack, I suggest that you end the Introduction by emphasizing the specific contributions of this work with respect to others in a more evident way.
The conclusion section is an appropriate place to list these highlights too.
